Temporal comparison and predictors of fish species abundance and richness on undisturbed coral reef patches

Wagner Elena L.E.S.
Roche Dominique G.
Binning Sandra A. binningsandra@gmail.com
Wismer Sharon
Bshary Redouan
Institute of Biology, University of Neuchâtel , Neuchâtel, NE , Switzerland
Esteban María Ángeles
Electronic publication date: 2015 Dec 1
Publication date: 2015
Volume: 3
Electronic Location ID: e1459
Received 2015 Aug 5; Accepted 2015 Nov 10
Copyright: © 2015 Wagner et al.
Copyright year: 2015
Copyright holder: Wagner et al.
License: This is an open access article distributed under the terms of the Creative Commons Attribution License, which permits unrestricted use, distribution, reproduction and adaptation in any medium and for any purpose provided that it is properly attributed. For attribution, the original author(s), title, publication source (PeerJ) and either DOI or URL of the article must be cited.
License URL: https://creativecommons.org/licenses/by/4.0/

Keywords: Cleaner wrasse, Coral reef fishes, Labroides dimidiatus, Red Sea, Long-term monitoring, Stability, Refuges

Funding: Swiss Science Foundation Fonds de Recherches du Québec: Nature et Technologies The study was supported by grants from the Swiss Science Foundation (RB) and the Fonds de Recherches du Québec: Nature et Technologies (SAB, DGR). The funders had no role in study design, data collection and analysis, decision to publish, or preparation of the manuscript.

==============================
Large disturbances can cause rapid degradation of coral reef communities, but what baseline changes in species assemblages occur on undisturbed reefs through time? We surveyed live coral cover, reef fish abundance and fish species richness in 1997 and again in 2007 on 47 fringing patch reefs of varying size and depth at Mersa Bareika, Ras Mohammed National Park, Egypt. No major human or natural disturbance event occurred between these two survey periods in this remote protected area. In the absence of large disturbances, we found that live coral cover, reef fish abundance and fish species richness did not differ in 1997 compared to 2007. Fish abundance and species richness on patches was largely related to the presence of shelters (caves and/or holes), live coral cover and patch size (volume). The presence of the ectoparasite-eating cleaner wrasse, Labroides dimidiatus, was also positively related to fish species richness. Our results underscore the importance of physical reef characteristics, such as patch size and shelter availability, in addition to biotic characteristics, such as live coral cover and cleaner wrasse abundance, in supporting reef fish species richness and abundance through time in a relatively undisturbed and understudied region.

Introduction

Coral reefs and the biodiversity they support are declining globally as a result of natural and anthropogenic stressors (e.g., Bellwood et al., 2004; Gardner et al., 2005; De’ath et al., 2012). Phenomena such as climatic fluctuations, disease outbreaks, species invasions and severe storms can negatively impact coral reef species’ abundance and diversity, and are being exacerbated by human activities such as overfishing, pollution and tourism (Hughes, 1994; Hughes & Connell, 1999; Pandolfi et al., 2003; Gardner et al., 2005; Hasler & Ott, 2008; Hoegh-Guldberg & Bruno, 2010; De’ath et al., 2012). To better predict the long-term impacts of such disturbances on reef communities, it is critical to understand how these communities change through time in the absence of large perturbations, especially in relatively pristine areas which have not been heavily impacted by tourism or human development (Knowlton & Jackson, 2008; Edmunds et al., 2014; Graham et al., 2014; Beldade et al., 2015). This information provides a valuable baseline against which changes caused by disturbances can be assessed.

Long-term monitoring of coral reefs is also essential for understanding the factors promoting reef resilience, stability and recovery, as well as identifying and managing threats (Hughes et al., 2010; De’ath et al., 2012; Edmunds et al., 2014; Alvarez-Filip et al., 2015). Some successful long-term monitoring programs exist in well-studied regions such as the Hawaiian archipelago, the Great Barrier Reef and the Caribbean (i.e., Hughes & Connell, 1999; Gardner et al., 2005; De’ath et al., 2012; Edmunds et al., 2014; Alvarez-Filip et al., 2015). However, we still lack temporal data for lesser-studied regions in the Indo-Pacific and the Red Sea (Berumen et al., 2013, but see Edmunds et al., 2014; McClanahan et al., 2014; Beldade et al., 2015 for recent comparisons in French Polynesia and the Western Indian Ocean). Data from these lesser-studied regions will help fill crucial gaps in our understanding of the global patterns driving coral reef decline.

Several physical properties of reef habitats are associated with increased resilience and/or biodiversity including reef zone, depth, area, structural complexity and shelter availability (e.g., Bellwood & Hughes, 2001; Ménard et al., 2012; Graham & Nash, 2013; Graham et al., 2014; Graham et al., 2015). In addition, biotic factors such as live coral cover and the presence/absence of key functional groups or organisms are important determinants of fish abundance and diversity (e.g., Bshary, 2003; Grutter, Murphy & Choat, 2003; Jones et al., 2004). Specifically, the long-term presence of the ectoparasite-eating cleaner wrasse, Labroides dimidiatus, is associated with increased growth, condition and abundance of its “client” fishes (Grutter, Murphy & Choat, 2003; Clague et al., 2011; Waldie et al., 2011; Bonaldo, Hoey & Bellwood, 2014, Fig. 1). However, few studies explicitly include cleaner wrasse abundance alongside other biotic and abiotic factors when evaluating predictors of reef fish species richness and abundance.

Figure 1 Photo of a cleaning interaction.

The cleaner wrasse, Labroides dimidiatus, removes ectoparasites from the surface of a ‘client’ reef fish (Photo: L. dimidiatus cleaning an anthias, Pseudanthias squamipinnis, Red Sea, S. Gingins).

We surveyed live coral cover and coral reef fishes in 1997 and again in 2007 on 47 patch reefs at Mersa Bareika, Ras Mohammed National Park, Egypt, to (1) assess natural changes in coral cover and fish species abundance and richness in an undisturbed reef system between two points in time and (2) identify important biotic and abiotic predictors of fish abundance and species richness using reef patches as replicates.

Materials and Methods

Study site

Ras Mohammed National Park is located at the southern tip of the Sinai Peninsula in the Red Sea, Egypt (Fig. 2). Established as a protected area in 1983, the park spans an area of 480 km2 including coral reef, coastline, mangrove and desert habitats (SEAM, 2005). Threats to the area include shipping traffic, oil spills, invasive species (e.g., the crown-of-thorns starfish, Acanthaster planci), climate change (e.g., increased temperatures causing coral bleaching), and tourism (Ormond et al., 1997; SEAM, 2005; Tilot et al., 2008). However, Mersa Bareika (27°47′23.9″N, 34°13′02.8″E), located inside the Ras Mohammed National Park, is a designated research area, which restricts human activity occurring in the region and was closed to tourists during the study period. No natural or human disturbance is known to have occurred in the study area during the ten years between surveys. An outbreak of A. planci in 1998 affected some of the reefs in Ras Mohammed National Park. However, the Egyptian Environmental Affairs Agency reacted quickly to the threat, thus minimizing the impact at this site (PERSGA/GEF, 2003). No A. planci were observed on the patch reefs in Mersa Bareika throughout the outbreak during surveys by RB in 1998. Similarly, the 1998 global coral bleaching event, which caused substantial damage in the southern Red Sea and Gulf of Aden, did not affect the northern Red Sea (Kobt et al., 2004). Permission to carry out research in Mersa Bareika was granted to RB by the Egyptian Environmental Affairs Agency, locally verified by the authorities at Ras Mohammed National Park.

Habitat surveys

Mersa Bareika is a narrow fringing reef along the coast of a well-protected bay (Fig. 2). Sand enters the water through wadis channels that flood during the rainy season, creating small patch reefs separated by at least 5 m of sand. Forty seven patch reefs (depth 0.68 m to 7.57 m) spanning 450 m of shoreline were surveyed in November 1997 (see Bshary, 2003) and again between September and November 2007 (Fig. 2 and Table S1). Average depth below the surface was estimated for each patch with a dive computer. Patch volume was estimated following methods described in Bshary (2003). Maximum patch length, width, and average height above the substrate were used to calculate volume according the closest geometrical shape of the patch (e.g., rectangle, ellipse). Percentage live coral cover on each patch was estimated using a modified line-intercept method (English, Wilkinson & Baker, 1997): a transect tape was laid out across the longest section of each patch and coral cover estimated as the proportion of the transect tape that was laid over live coral. The tape was then set perpendicular to this axis, across the widest area of the patch, and the process repeated. The total percentage area of live coral cover was calculated as the mean of these two measures. The total percentage of holes and caves on each patch was also estimated in this way. Holes were defined as small cavities (typically 1–3 cm wide) on the surface of the reef matrix (excluding spaces in between coral). Caves were defined as openings at the base of the reef where the reef matrix connected to the substrate.

Figure 2 Study site map.

Map of the patch reefs and study area at Mersa Bareika, Ras Mohammed National Park, Egypt (27°47′23.9″N, 34°13′02.8″E). The dotted line defines the park boundaries.

Fish surveys

Coral reef fishes were categorized into four groups depending on their movement patterns and habitat use (Table S2; Bshary, 2001). Resident fishes have small home ranges restricted to a single patch reef and include damselfishes (Pomacentridae), cardinalfishes (Apogonidae), squirrelfishes (Holocentridae) and some wrasses (Labridae). Visitors have larger home ranges that overlap several patches, and include parrotfishes (Scaridae), goatfishes (Mullidae) and fusiliers (Caesionidae). Butterflyfishes (Chaetodontidae) remain on one patch for long periods (hours to days) but also move between patches, and were considered facultative visitors (i.e., intermediate home ranges). Species that we could not classify into any of the three categories were categorized as undetermined (e.g., Cyrrithidae, Tetraodontidae).

Reef fish abundance and species richness were surveyed by two SCUBA divers three times in 1997 and five times in 2007, at two week intervals. One observer identified species and recorded their abundance at each patch. The second diver also recorded species’ identity to ensure the primary observer had not missed a species. Upon arrival at a patch, observers recorded fishes in the following order: visitors, facultative visitors, residents and undetermined. The abundance of species occurring in shoals or schools was estimated in multiples of ten. Fishes that swam to patches after the count had begun were not recorded. Cryptic species including most blennies (Blenniidae) and gobies (Gobiidae) were excluded from the surveys except for the scale-eating blenny (Plagiotremus tapeinosoma), the bluestriped blenny (Plagiotremus rhinorhynchos), the mimic blenny (Aspidontus taeniatus), and the blackfin dartfish (Ptereleotris microlepis). Species that primarily occupy sandy areas such as pipefishes (Syngnathidae) and lizardfishes (Synodontidae) were also excluded.

Statistical analyses

We examined differences in percent live coral cover between 1997 and 2007 on the 47 patches using a paired t-test. Percent live coral cover was logit transformed (Warton & Hui, 2010). We used the Information Theoretic approach (Burnham & Anderson, 2002; Johnson & Omland, 2004; Grueber et al., 2011) to determine which variables best explained variation in fish abundance (number of individuals) and species richness (number of species) among patch reefs and years. Response variables were averaged across fish surveys within year. Therefore, the data were not Poisson distributed or zero inflated. Two full models were constructed using general linear mixed-effects models (LMM; lmer function in R). Model assumptions were checked with plots of residuals vs. fitted values and qqplots of residuals for fixed and random effects. Fish abundance was log transformed to meet model assumptions. Seven predictor variables were included in each model: survey year, depth below surface, patch volume, percent live coral cover, percent caves, percent holes and cleaner wrasse abundance. Cleaner wrasse abundance was not included in the fish abundance counts. We graphically examined all two-way interactions and excluded those that had no apparent effect on the response variables for model simplicity and parsimony following recommendations in Burnham & Anderson (2002). We examined non-linear relationships using pairs plots and plots of generalized additive models (GAM). Quadratic terms were included where necessary. Patch ID was included as a random factor in both models. Continuous predictors were z-standardized (mean = 0, SD = 1) and a correlation plot used to check for collinearity. All correlation coefficients were <0.35. See Tables S3 and S4 for details of the main effects and interaction terms included in the full models.

We used the Akaike Information Criterion modified for small sample sizes (AICc) to select the best candidate models for each response variable with the “dredge” function in the R package MuMIn. We performed model averaging using the “model.avg” function (MuMIn) when the normalized Akaike weight value (wim) of the best model was <0.9 (Burnham & Anderson, 2002). The confidence set of candidate models selected for model averaging included all models for which the wim fell within 10% of the maximum normalized weight, suggesting that these models had substantial support in explaining the data (Burnham & Anderson, 2002). We calculated the marginal R2 (variance explained by the fixed factors; RGLMMm2) and conditional R2 (variance explained by the fixed and random factors; RGLMMc2) following Nakagawa & Schielzeth (2013). All analyses were done in R 3.1.2 (R Development Core Team , 2014). The data and code for this study are deposited in the public respository figshare (Wagner et al., 2015) following best practices (White et al., 2013; Roche et al., 2015).

Results and Discussion

There was no difference in percent live coral cover on the 47 patches surveyed in 1997 and 2007 (t46 = 1.48, p = 0.15; Table 1). This pattern was not driven by a shift to soft corals as documented in Tilot et al. (2008): no information is available for 1997, but soft corals made up only a small proportion of live corals in 2007, i.e., 0–10% on 41 out of 47 patches.

Table 1 Comparison of patch characteristics between years.

Mean and mean paired difference (±one standard error) for percent live coral cover, fish abundance and fish species richness on 47 patch reefs at Ras Mohammed National Park, Egypt, in 1997 and in 2007.

Year	Coral cover	Abundance	log(abundance)	Richness	
2007	22.5 ± 1.8	109.9 ± 15.9	1.86 ± 0.06	20.5 ± 1.3	
1997	24.9 ± 1.8	147.7 ± 22.2	1.96 ± 0.07	18.6 ± 1.1	
Paired difference	−2.4 ± 2.0	−37.9 ± 12.8	−0.10 ± 0.04	2.0 ± 0.6	

Eighteen and thirty models had support in explaining fish abundance and species richness on patch reefs based on AICc scores, respectively (Table 2, Tables S3 and S4). In both cases, fixed factors explained over 60% of the variance; this percentage exceeded 80% when accounting for both fixed and random factors (Table 2).

Table 2 Model selection results.

Predictors and interaction terms included in the best models explaining variation in (A) fish abundance and (B) species richness among patch reefs (n = 47) and years (1997 vs. 2007). The model averaged parameters estimates (β), unconditional standard errors (SE), 95% confidence interval (95% CI), and the normalized Akaike weight (wip) for each predictor are shown. Also included is the number of models (Models) in which predictors were included and marginal (RGLMMm2) and conditional (RGLMMc2) R2 values for the mixed model (LMM) with predictors identified as important. Predictors are in order of importance (wip); those for which the 95% CI does not overlap zero are indicated in bold. All models include a constant. See Supplemental Information 1 for detailed tables.

Predictor	β	SE	95% CI	wip	Models	
(A) Fish abundance	
Intercept	4.85	0.15	4.56–5.15	1.00	18	
Percent caves (PC)	0.35	0.12	0.12–0.58	1.00	18	
Percent live coral cover (PLCC)	0.23	0.06	0.11–0.36	1.00	18	
Patch volume (PV)	1.05	0.14	0.78–1.32	1.00	18	
PV2	−0.42	0.11	−0.63–−0.21	1.00	18	
PC:PLCC	0.21	0.06	0.08–0.33	1.00	18	
PC:PV	0.41	0.13	0.16–0.66	1.00	18	
L. dimidiatus abundance (Ldim)	0.10	0.06	−0.01–0.22	0.63	11	
Year	−0.16	0.09	−0.33–0.01	0.57	9	
Percent Holes (PH)	−0.03	0.08	−0.18–0.13	0.31	10	
Depth	−0.02	0.08	−0.18–0.15	0.15	4	
PH:Ldim	0.08	0.05	−0.02–0.17	0.09	3	
PC:PH	0.09	0.09	−0.09–0.28	0.08	4	
RGLMMm2=0.64, RGLMMc2=0.82	
(B) Fish species richness	
Intercept	20.44	0.98	18.51–22.37	1.00	30	
Percent live coral cover (PLCC)	1.23	0.41	0.44–2.03	1.00	30	
L. dimidiatus abundance (Ldim)	1.75	0.39	0.99–2.52	1.00	30	
Year	2.87	0.59	1.71–4.03	1.00	30	
Patch volume (PV)	8.34	0.92	6.53–10.15	1.00	30	
PV2	−2.48	0.76	−3.97–−1.00	1.00	30	
Percent caves (PC)	1.68	0.78	0.15–3.20	0.96	29	
PC:PV	2.21	0.82	0.61–3.81	0.91	26	
PC:PLCC	0.88	0.40	0.10–1.67	0.77	21	
Depth	−0.77	0.54	−1.83–0.29	0.70	23	
Percent holes (PH)	0.71	0.53	−0.33–1.74	0.62	18	
PH:PV	1.60	0.65	0.33–2.87	0.55	14	
PLCC:Depth	0.97	0.45	0.09–1.85	0.48	15	
PC:Ldim	0.69	0.70	−0.68–2.05	0.21	9	
PH:Ldim	0.20	0.33	−0.47–0.84	0.11	5	
RGLMMm2=0.81, RGLMMc2=0.90	

We found no marked difference in fish abundance in 1997 and 2007 (Tables 1 and 2A). The presence of caves and live coral cover were strongly, positively related to fish abundance, and the effect of live coral cover was stronger when the presence of caves was high (Fig. S1). Patch volume also had a positive (saturating) effect on fish abundance but only when caves were abundant (Fig. S2). These findings support previous studies which stress the importance of shelters and live coral in positively influencing fish abundance (Jones et al., 2004; Ménard et al., 2012; Graham & Nash, 2013). We also found a positive correlation between fish abundance and cleaner wrasse abundance, although the confidence interval of this estimate overlapped zero (Table 2A).

Across all patches, a total of 86 species from 24 fish families were recorded in 1997, and 118 species from 29 fish families in 2007 (Table S2). This overall trend was reflected at the patch level by a slight increase in fish species richness between years (Tables 1 and 2B). Patch species richness was positively related to live coral cover, caves, holes, patch volume, and the presence of cleaner fish (Table 2B and Fig. S3). Similar to fish abundance, the relationship between live coral cover and species richness was stronger when more caves were present, and the positive (saturating) effect of patch size (volume) was greater when caves were abundant (Fig. S4). Patch volume was also positively related to species richness and the strength of this relationship increased as the presence of holes increased (Fig. S5). Live coral cover was more strongly related to species richness as depth increased (Fig. S6) perhaps as a result of an increase in the number of resident and/or coral-dependent species at greater depths. Importantly, cleaner fish abundance was as important as live coral cover, caves and patch volume at predicting fish species richness on patches (Table 2B).

Recent studies have identified a causal link between cleaner wrasse presence and client fish diversity. Short and long-term experimental removal of L. dimidiatus on patch reefs in Egypt and Australia led to decreases in mean fish size, species abundance and/or richness of fishes on reefs without cleaners (Bshary, 2003; Waldie et al., 2011). Our results support these findings, and suggest that natural variation in cleaner wrasse abundance on reefs is also related to fish species richness and, to a lesser extent, abundance. The strong effect of cleaner wrasse on coral reef fish communities is remarkable given that these cleaners are small (<15 cm) and uncommon relative to other reef dwellers, suggesting that positive interactions among reef fish species are important, but often overlooked, components of reef communities (Grutter & Irving, 2007).

Despite increasing protection and awareness of the need to conserve coral reef habitats, studies suggest that the global decline in coral reef health has been rapid, severe and may be irreversible (Hughes, 1994; Veron et al., 2009; Hoegh-Guldberg & Bruno, 2010). Yet, some near-pristine reef habitats remain, and temporal monitoring of these undisturbed areas provides crucial baselines, which can help inform conservation planning and management of degraded reefs (Knowlton & Jackson, 2008). Our results provide crucial data on the status of a series of patch reefs in the Red Sea, a region that remains understudied despite a well-developed tourism industry (Berumen et al., 2013). Our temporal comparison revealed that undisturbed patch reefs in the Red Sea had similar values of live coral cover and fish species richness and abundance in 1997 as in 2007 according to our point-in-time sampling. Our results also underscore the importance of physical reef characteristics, such as patch size and shelter availability, in addition to biotic characteristics, such as live coral cover and cleaner wrasse abundance, in supporting reef fish assemblages. These reef properties should thus be considered in future management plans aimed at promoting increases in coral reef fish abundance and species richness.

Supplemental Information

Supplemental Information 1 Supplemental Tables and Figures

Tables S1, S2, S3, S4 Figures S1, S2, S3, S4, S5, S6.

Click here for additional data file.

We thank the authorities at Ras Mohammed National Park and the Egyptian Environmental Affairs Agency for permission to conduct this research, L Chapuis, I Riepl, Abdallah, Raffia and Hamoo for field and logistic support, R Slobodeanu and K Turgeon for statistical advice, S Meyer, C Strübin, A Pinto, and R Bergmüller for helpful discussion, and three anonymous reviews for comments on a previous version of this manuscript.

Additional Information and Declarations

Competing Interests

Author Contributions

Animal Ethics

Field Study Permissions

Data Availability

The authors declare there are no competing interests.

Elena L.E.S. Wagner performed the experiments, contributed reagents/materials/analysis tools, wrote the paper, prepared figures and/or tables, reviewed drafts of the paper.

Dominique G. Roche and Sandra A. Binning analyzed the data, contributed reagents/materials/analysis tools, wrote the paper, prepared figures and/or tables, reviewed drafts of the paper.

Sharon Wismer contributed reagents/materials/analysis tools, prepared figures and/or tables, reviewed drafts of the paper.

Redouan Bshary conceived and designed the experiments, performed the experiments, contributed reagents/materials/analysis tools, reviewed drafts of the paper.

The following information was supplied relating to ethical approvals (i.e., approving body and any reference numbers):

This was an observational study, and did not require approval from an animal ethics board in Switzerland or Egypt. No animals were harmed during this study.

The following information was supplied relating to field study approvals (i.e., approving body and any reference numbers):

Permission to carry out research in Mersa Bareika was granted by the Egyptian Environmental Affairs Agency (EEAA) and locally enforced at entrance checkpoints by the authorities at Ras Mohammed National Park. In 1996, Redouan Bshary applied in writing to the EEAA, and he and his students were thus added to the list of scientific researchers permitted to enter the park for research purposes during this year. This process was repeated in 2007.

The following information was supplied regarding data availability:

Data and code are deposited in the public respository figshare (DOI: 10.6084/m9.figshare.1335775; Wagner et al., 2015).

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
