# Peer review of "Temporal comparison and predictors of fish species abundance and richness on undisturbed coral reef patches"

_PeerJ, doi:10.7717/peerj.1459_

## Round 0.1 · original submission · Minor Revisions

Please, consider all the suggestions in the revised manuscript.

Reviewer 1 ·

Basic reporting

1. Line 34. Should be ‘intensity’ not ‘intensify’
2. Is there a particular reason most (but not all) of the Family names are in italics?

Experimental design

1. The introduction of the manuscript expresses an aim to better understand coral reef community responses to disturbances (natural/anthropogenic) by gaining a better understanding of changes to the reef community on relatively undisturbed reefs. I believe the authors could have achieved the aims of their paper much more informatively if they’d used their assemblage-level data. Currently, the very simple metrics they use (live coral cover, fish abundance, fish species richness) give only a very basic and superficial assessment of their study site.

Validity of the findings

1. A well-executed series of surveys of a relatively undisturbed area of coral reefs, supporting the general consensus that reef complexity and live coral cover are important to reef fish abundance and species richness.

Additional comments

I have previously reviewed this paper for another journal and am happy to see that several earlier recommendations have been incorporated into this version of the manuscript.

Reviewer 2 ·

Basic reporting

Well written, no extraneous information.

A few relatively minor comments

(i) I think you mean ‘tourism intensity’ not ‘tourism intensify’ on line 34
(ii) Lines 46-53 seem to me to overstate the lack of long-term monitoring data from many regions. It’s arguable that the data are not as widely available as they might be, or that there are insufficient well studied and publicized time series, but there are certainly a lot of ongoing efforts, including at many remote locations - by NOAA in the Pacific, by the IRD group and CRIOBE in French Polynesia, and a lot of work in East Africa and the Indian Ocean by McClanahan and colleagues. I am sure there are also many other efforts that I am not so aware of.
(iii) I believe I do understand how the models were constructed and run, and how model averaging was done - and I believe the analysis is well done - but I think more information should be added to the methods section to explain the approach so that a reader does not have to glean that from scrutinizing the results. For example, I would recommend being explicit about the R package used and the procedure to run all possible models and then to do the model averaging (in the MuMin package that would presumably be the dredge and model.avg routines). I assume also that all models were included until cumulative model weight was >0.9 [which incidentally seems like a lot of very unlikely models were included, although that is probably fine for model averaging]). However, all that is not entirely clear to me from how the methods are written now. I recommend authors including simplified versions of their analysis scripts in the supplementary materials, but that is not essential.
(v) Figures in supplementary materials are very helpful. I think they would be even better if the figures could be generated for untransformed response and predictor values, so that true shape of the relationship between predictors and response variable was clear. There could also be an argument for adding more figures (e.g. to show relationship between richness and cleaner wrasse density) as the shape and slope of the relationships is usually very interesting.

Experimental design

Excellent

Validity of the findings

The one thing I quibble with is the statement in several places (line 20, line 168-169) that coral cover or patch reefs remained relatively ‘unchanged’ over the 10 year time period. There is only data from two time points 1997 and 2007, as such we can not know whether there has been change over that time .. we only know that the start and end points are relatively similar.

---

## Round 0.2 · accepted · Accept

The authors have improved the revised manuscript according to the reviewer's suggestions.